# TALK UNTIL YOU BURN OUT: ESCALATING 3D-LLM OVERGENERATION VIA SEMANTIC MANIPULATION

## ABSTRACT

The rise of 3D large language models (3D-LLMs) has unlocked new potential in multimodal reasoning over unstructured 3D data, powering applications such as robotics and autonomous driving. However, these models also face new security risks, particularly during the inference-time computation. In this work, we present **Exhaust3D**, the first targeted energy-oriented adversarial framework against 3D-LLMs. Exhaust3D performs a **resource exhaustion attack** by injecting imperceptible yet strategically structured semantic perturbations into 3D point clouds, causing the model to overgenerate outputs and inflate inference latency. Specifically, we design two key components: (1) a *semantic-aware adversarial manipulation strategy* that leverages internal model representations to selectively perturb semantically critical point regions while preserving geometric structure, and (2) a *trajectory disruption mechanism* that maintains high-entropy token predictions to prolong auto-regressive decoding and induce verbose outputs. Experiments on widely-used 3D-LLM benchmarks show that Exhaust3D increases decoding steps and energy consumption by up to **6.45×** with negligible degradation in functional performance. These results expose a previously underestimated vulnerability of 3D-LLMs to resource exhaustion attacks, highlighting the urgent need for energy-aware robustness in future multimodal foundation models.

## 1 INTRODUCTION

Recent advances in 3D large language models (3D-LLMs) such as PointLLM (Xu et al., 2024; 2025)and X-InstructBLIP (Panagopoulou et al., 2023) have significantly improved spatial-language reasoning, enabling robust performance in a wide range of 3D tasks including object understanding, scene-level decision-making, and multimodal navigation. These models typically adopt a language foundation model (LLM) as their backbone and extend it to the 3D domain by aligning textual and geometric modalities, thereby generalizing the LLM's linguistic capabilities to visual and spatial perception. Consequently, 3D-LLMs are increasingly deployed in high-stakes applications such as autonomous driving, embodied AI, and digital reconstruction.

However, the powerful generative and reasoning capabilities of 3D-LLMs come at the cost of enormous model sizes and expensive inference pipelines. In particular, their autoregressive decoding requires substantial computational resources per query. Moreover, due to the limited availability and high cost of constructing 3D datasets, users often rely on publicly available 3D assets (e.g., meshes, point clouds) sourced from the internet to support downstream reasoning tasks. This opens up a new vulnerability: an adversary may subtly manipulate these 3D inputs to induce 3D-LLMs to generate abnormally long outputs at inference time—causing excessive energy usage or cloud token consumption, thereby launching a resource exhaustion attack.

Several prior works have explored resource exhaustion attacks in text and image modalities. In the text domain, Sponge examples (Shumailov et al., 2021) increase inference cost by maximizing activation norms across layers. Meanwhile, Engorgio Prompt (Dong et al., 2024) demonstrates that carefully crafted prompts can suppress end-of-sequence (EOS) token generation in LLMs, forcing abnormally long outputs without degrading semantic quality. In the vision-language setting, NICGSlowdown (Chen et al., 2022) manipulates logit dynamics to delay EOS emission, and Verbose Images (Gao et al., 2024) introduce imperceptible perturbations that promote diverse outputs in VLMs, thereby amplifying computational overhead.

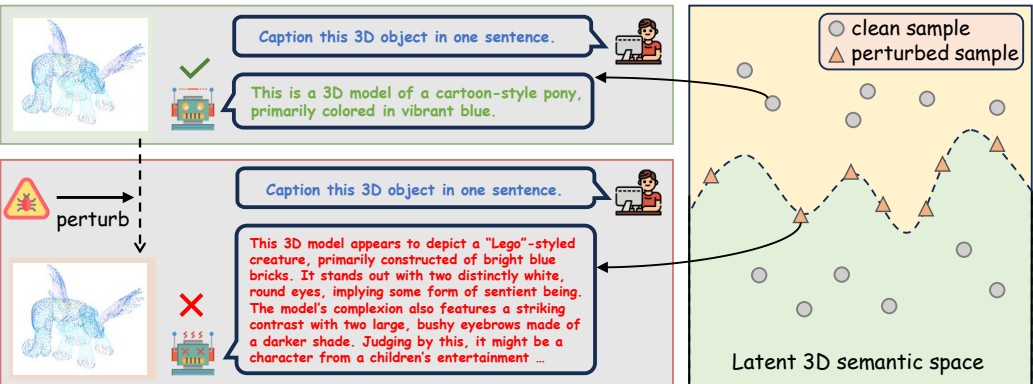

Figure 1: Demonstration of Exhaust3D's effects. **Left (output level):** Adversarial perturbations significantly prolong auto-regressive decoding while preserving input fidelity. **Right (model level):** Perturbations shift samples into regions of semantic ambiguity, blurring distinctions from clean examples in semantic space.

Despite these efforts, existing computation-based attacks have primarily focused on LLMs and VLMs, whereas the study of 3D-LLMs remains largely unexplored. The unique characteristics of 3D domains pose distinctive challenges for effective attacks, such as maintaining geometric fidelity, ensuring robust cross-modal alignment, and addressing the semantic sparsity inherent in 3D spaces. To bridge this gap, we present **Exhaust3D**, the first adversarial framework that performs semantic-aware resource exhaustion attacks against 3D-LLMs through purely 3D-modal perturbations. Exhaust3D injects imperceptible but strategically crafted changes into point cloud inputs to manipulate the LLM's generation process and escalate inference-time energy consumption. The key components of our method are two-fold: (1) **Semantic-aware adversarial manipulation**: We leverage the model's internal hidden states to identify semantically important tokens and selectively perturb their corresponding points in the input space, disrupting the model's reasoning process while preserving the overall geometric structure. (2) **Trajectory disruption mechanism**: We maintain high-entropy token predictions and suppress premature EOS emission to prolong auto-regressive decoding, triggering verbose outputs and significantly increasing inference-time energy consumption. Figure 1 illustrates the effect of Exhaust3D. Experiments on widely-used 3D-LLM benchmarks show that Exhaust3D substantially increases decoding steps and energy consumption, exposing a critical vulnerability in 3D multimodal reasoning systems. By combining semantic-aware adversarial manipulation and a trajectory disruption mechanism, the attack induces excessively long sequences in a highly stealthy and resource-draining manner. Our contributions are summarized as follows:

- To the best of our knowledge, we propose the first attack framework named Exhaust3D, which performs semantic-aware **resource exhaustion attacks** on 3D-LLMs via imperceptible perturbations crafted entirely in the 3D input space.

- We design a novel ambiguity-driven trajectory disruption mechanism that effectively manipulates token dispersion and persistence to prolong decoding and induce verbose, energy-expensive outputs from various 3D-LLMs.

- We conduct comprehensive experiments on the Objaverse (Deitke et al., 2023) and Model-Net40 (Wu et al., 2015) benchmark, showing that Exhaust3D increases output length and energy cost by up to 6.45× and 6.12× respectively, demonstrating consistent and powerful attack performance.

## 2 RELATED WORK

### 2.1 RECENT ADVANCES IN 3D-LLMS

Recent advances in 3D large language models (3D-LLMs) have substantially enhanced the ability of language models to perceive and reason over 3D data. A first line of research leverages 2D vision-language models for 3D understanding. 3D-LLM (Hong et al., 2023) renders 3D objects into

multi-view images and applies CLIP-like encoders (Radford et al., 2021) together with BLIP (Li et al., 2023a) to bridge 3D perception and language understanding.

Beyond explicit multi-view rendering, another line of work focuses on directly aligning 3D point clouds with language models. Point-Bind (Guo et al., 2023) constructs a shared multimodal embedding space under the guidance of ImageBind (Girdhar et al., 2023), using image features to assist point cloud alignment and enabling generative capabilities through 2D multimodal models like ImageBind-LLM (Han et al., 2023). PointLLM (Xu et al., 2024; 2025) encodes 3D point clouds into latent point tokens and concatenates them with textual tokens for auto-regressive LLM processing, enabling end-to-end spatial-language reasoning from 3D input. LEO (**?**) extends this framework by incorporating additional modalities to enhance cross-modal reasoning for downstream tasks.

A number of other models explore advanced 3D reasoning and scene-level understanding. ShapeLLM (Qi et al., 2024) combines contrastive pretraining with cross-modal alignment to enable zero-shot generalization. MiniGPT-3D (Tang et al., 2024) improves scalability through a multi-stage projection pipeline from 3D point clouds to token representations. GreenPLM (Tang et al., 2025) achieves data-efficient learning by mapping point clouds into text space, requiring only 12% of 3D training data. Video-3D LLM (Zheng et al., 2025) introduces 3D positional encoding into video representations for reasoning over dynamic scenes. X-InstructBLIP (Panagopoulou et al., 2023) aligns multiple modalities with a frozen LLM using complementary Q-Former and projection modules. Finally, LSceneLLM (Zhi et al., 2025) targets large-scale scene understanding with adaptive preference identification and scene magnification.

## 2.2 RESOURCE EXHAUSTION ATTACKS

In the field of computer security, Denial-of-Service (DoS) attacks are a classic and pervasive threat. Denial-of-Service (DoS) attacks aim to exhaust system resources or bandwidth, preventing legitimate users from accessing services (Elleithy et al., 2005; Aldhyani & Alkahtani, 2023; Bhatia et al., 2018). Typical strategies include overwhelming servers with massive request floods or exploiting vulnerabilities (Mirkovic & Reiher, 2004; Long & Thomas, 2001). Such attacks pose long-standing risks to real-world platforms, as they directly undermine service availability.

With the emergence of adaptive neural architectures, including adaptive computation time networks and large-scale models such as Large Language Models (LLMs) and Vision-Language Models (VLMs), inference-time computation has become input-dependent. Unlike conventional networks with fixed costs, these models dynamically adjust decoding length or intermediate computation according to input content. This paradigm shift has enabled a new family of resource exhaustion attacks, which drain computational resources by inducing excessive energy consumption and latency through adversarial inputs (Hong et al., 2020; Liu et al., 2023). For example, sponge samples (Shumailov et al., 2021) maximize the L2 norm of intermediate activations to trigger redundant computation, while NICGSlowDown (Chen et al., 2022) manipulates end-of-sequence (EOS) logits to prolong sequence generation and increase decoding overhead.

In the era of generative models, resource exhaustion attacks have also emerged against LLMs and VLMs. Engorgio Prompt (Dong et al., 2024) shows that carefully crafted prompts can suppress EOS token generation in LLMs, forcing abnormally long outputs without degrading semantic quality. Verbose Images (Gao et al., 2024) extends this idea to vision-language models by delaying EOS generation, increasing output uncertainty, and enhancing token diversity to induce energy-intensive decoding. LLMEffiChecker (Feng et al., 2024) further identifies critical tokens and constructs input variants to systematically reduce LLM inference efficiency across layers. In the 3D domain, Poison-splat (Lu et al., 2024) perturbs multi-view data to substantially increase the computational overhead of 3D Gaussian Splatting (Kerbl et al., 2023), highlighting how input-level perturbations translate into heavy resource consumption.

Despite this growing body of work, **resource exhaustion attacks targeting 3D-LLMs remain largely unexplored**. These models combine multi-view perception, geometric reasoning, and cross-modal text generation, leading to complex and input-dependent inference pipelines. Such characteristics create new vulnerabilities but also raise unique challenges for attack design. Our work takes the first step toward systematically exploring and exploiting these vulnerabilities, providing insights that can guide the development of more robust and resource-efficient 3D-LLMs.

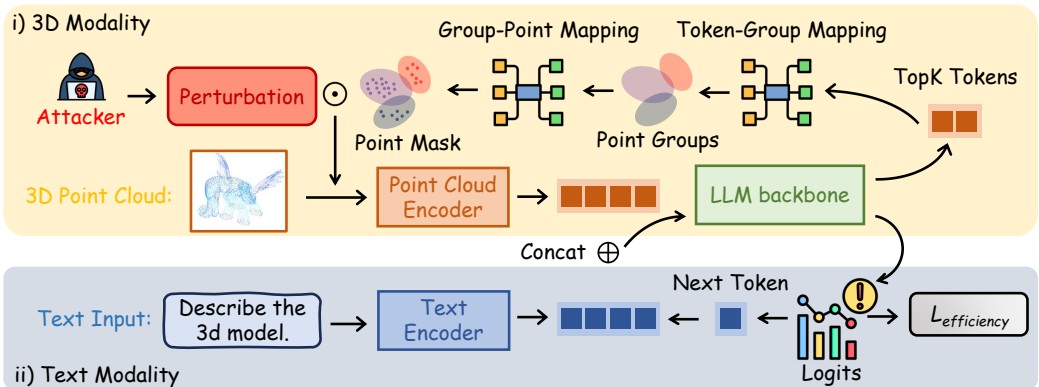

Figure 2: Overall pipeline of Exhaust3D. The input 3D point cloud $P$ is first tokenized, and subtle perturbations are injected only at semantically critical points indicated via the token–group–point mapping, creating targeted semantic ambiguity. Point tokens are then combined with textual embeddings to drive auto-regressive decoding in the LLM. By evaluating the predicted logits, the efficiency-oriented loss $\mathcal{L}_{\text{efficiency}}$ directs the perturbations to extend output sequences and amplify computational cost.

## 3 METHODOLOGY

### 3.1 METHOD OVERVIEW

To extend generated sequences while keeping perturbations imperceptible, **Exhaust3D** leverages 3D point cloud structure through two modules. **Semantic-aware Adversarial Manipulation** identifies semantically critical points based on token importance and selectively perturbs them to preserve local geometry. **Trajectory Disruption Mechanism** prolongs generated outputs using a *dispersion loss* to increase token uncertainty and a *persistence loss* to delay EOS emission. To balance two losses, we adopt a projection-based adjustment optimization method. Together, these components generate longer outputs with minimal geometric distortion, amplifying the inference-time cost of 3D-LLMs. An overview of the attack pipeline is shown in Figure. 2.

### 3.2 PROBLEM FORMULATION

**3D-LLM Inference Procedure.** We consider a 3D large language model (3D-LLM) composed of an encoder and an auto-regressive decoder. Let $P \in \mathbb{R}^{N \times 3}$ denote the input point cloud and $T = \{t_1, \ldots, t_m\}$ the textual instruction. The encoder maps $P$ into latent point tokens $H_p$ and the textual input into tokens $H_t$. These are concatenated to form the initial context for the decoder. The decoder then generates an output sequence $Y = \{y_1, \ldots, y_k\}$ auto-regressively. At decoding step $s$, the model predicts token $y_s$ conditioned on the encoded inputs and the prefix $Y_{<s} = \{y_1, \ldots, y_{s-1}\}$. Concretely, the decoder logits at step $s$ are:

$$\mathbf{z}_s(P, T, y_{<s}) = f_{\text{dec}}([H_p; H_t], y_{<s}), \tag{1}$$

and the conditional probability of generating token $v \in V$ at step $s$ is:

$$p(v \mid P, T, y_{<s}) = \text{softmax}(\mathbf{z}_s(P, T, y_{<s}))_v. \tag{2}$$

**Attack Objective.** We aim to craft an imperceptible perturbation $\delta$ applied to the point cloud $P$ to obtain a perturbed input $\tilde{P} = P + \delta$, such that the 3D-LLM generates abnormally long output sequences $Y$, increasing inference-time computation. This is formalized by minimizing an efficiency-oriented objective:

$$\min_{\|\delta\|_\infty \leq \epsilon} \mathcal{L}_{\text{efficiency}}\left(f_{\text{dec}}([f_{\text{enc}}(P + \delta); H_t])\right), \tag{3}$$

where $\epsilon$ bounds the perturbation magnitude. The perturbation $\delta$ should preserve the geometric integrity of the original point cloud (See details in Section 3.3) while the minimization of $\mathcal{L}_{\text{efficiency}}$

aims to reduce the decoding efficiency, encourages high energy consumption during inference (See details in Section 3.4).

### 3.3 SEMANTIC-AWARE ADVERSARIAL MANIPULATION

To preserve the geometric fidelity of the input, we introduce a semantic-aware adversarial manipulation strategy that perturbs only a subset of semantically critical points. This strategy consists of three steps: (1) estimating token importance, (2) mapping tokens back to their corresponding points, and (3) applying masked perturbations under perceptual constraints.

We first measure the importance of each point token using the $\ell_2$ norm of its final-layer hidden state. The top-$k$ tokens are then selected according to a ratio $\rho$:

$$\mathcal{I} = \text{TopK}\big(\{\|\mathbf{h}_j^{(L)}\|_2\}_{j=1}^n, \; k = \lfloor \rho n \rfloor\big). \tag{4}$$

This step highlights the tokens with the strongest influence on the next-token logits and therefore the model's generation behavior. Each selected token $j$ corresponds to a local group of raw points $\mathcal{G}_j$, as determined by the encoder's grouping stage. We expand the token set $\mathcal{I}$ into a point-level set $\mathcal{P} = \bigcup_{j \in \mathcal{I}} \mathcal{G}_j$, which specifies all candidate points eligible for perturbation. In practice, we construct a **semantic-aware mask** $M \in \{0,1\}^N$ to mark these points, ensuring that gradients outside $\mathcal{P}$ are suppressed during optimization. Finally, the adversarial perturbation is applied in masked form:

$$\tilde{P} = P + \delta \odot M, \tag{5}$$

where $\delta$ denotes the additive perturbation constrained by an $\ell_\infty$ budget to maintain imperceptibility. During optimization, only masked points are updated, focusing perturbations on semantically influential regions while leaving the majority of the point cloud untouched. This semantic-aware mask design enables strong attack effectiveness with minimal perceptual distortion.

### 3.4 TRAJECTORY DISRUPTION MECHANISM

Building upon the semantic-aware perturbation strategy described above to maintain imperceptibility, we further introduce our trajectory disruption mechanism. It aims to prolong the auto-regressive decoding trajectory, which maximizes the inference-time energy cost of 3D-LLMs.

**From Adversarial Examples to Semantic Ambiguity.** In classification tasks, adversarial examples (AEs) are known to probe the decision boundary by applying small perturbations that induce misclassification (He et al., 2018; Ilyas et al., 2019; Rice et al., 2020). However, large language models, particularly 3D-LLMs, operate in an open-ended generative setting rather than a closed label space. In this context, the notion of a decision boundary must be reinterpreted. Instead of forcing high-confidence misclassifications, we transfer the functionality of AEs into the semantic space of generative models: perturbations are designed to push samples toward regions of high uncertainty, where token predictions become more ambiguous.

**Definition of SAI.** Formally, let $\tilde{P} = P + \delta$ denote the perturbed point cloud, and let $T$ denote the input text. Let $\hat{\mathbf{z}}^i \in \mathbb{R}^{|V|}$ denote the model's logits for the $i$-th token, and define

$$\hat{p}_i(\cdot) := \text{softmax}(\hat{\mathbf{z}}^i), \tag{6}$$

where $\hat{p}_i(v)$ gives the predicted probability for token $v \in V$. We call $\tilde{P}$ a **spatially ambiguous instance (SAI)** if its predictive distribution exhibits sufficiently high entropy:

$$\mathcal{H}(\hat{p}_i) \geq \log|V| - \tau, \tag{7}$$

where $\mathcal{H}(\hat{p}_i) = -\sum_{v \in V} \hat{p}_i(v) \log \hat{p}_i(v)$, $\log|V|$ is the entropy of the uniform distribution over the vocabulary, and $\tau \geq 0$ is a slack threshold. Intuitively, a smaller $\tau$ enforces a distribution closer to uniform (higher ambiguity); in the extreme $\tau = 0$ the condition requires near-uniformity. SAIs therefore correspond to perturbed samples that lie near the semantic boundary of the model's latent space, producing ambiguous predictions that prolong auto-regressive decoding and inflate inference-time energy consumption.

**Dispersion loss.** This geometric interpretation provides intuition for why SAIs exhibit unstable decoding behavior, and it naturally motivates the need for an objective that can deliberately induce such high-entropy states. To actively construct SAIs, we define a dispersion loss that encourages the predicted distributions to approach the maximum entropy distribution across the full sequence:

$$\mathcal{L}_{\text{dispersion}} = \mathbb{E}_{i=1}^{L}\Big[\mathrm{KL}(\hat{p}_i \,\|\, \mathcal{U})\Big], \tag{8}$$

where $\mathcal{U}$ is the uniform distribution over the vocabulary $V$, and the expectation is taken over sequence positions $i = 1, \ldots, L$. Minimizing $\mathcal{L}_{\text{dispersion}}$ drives the predicted distributions toward high entropy, thereby encouraging SAIs as defined in Eq. 7.

**Persistence loss.** Entropy maximization alone only diversifies token predictions but leaves the stopping behavior uncontrolled. To regulate the termination dynamics, we design a persistence loss that discourages premature EOS emission:

$$\mathcal{L}_{\text{persistence}} = \mathbb{E}_{i=1}^{L}\big[ -\log\big(1 - \hat{p}_i(\text{EOS})\big)\big], \tag{9}$$

where $\hat{p}_i(\text{EOS})$ denotes the predicted EOS probability at position $i$. This penalizes early EOS predictions and enforces a more persistent decoding trajectory. By combining the dispersion and persistence terms, we obtain the overall efficiency-oriented objective:

$$\mathcal{L}_{\text{efficiency}} = \mathcal{L}_{\text{dispersion}} + \mathcal{L}_{\text{persistence}}. \tag{10}$$

This joint formulation drives predictions toward high-entropy distributions while simultaneously suppressing EOS emission, thereby prolonging auto-regressive trajectories and inflating inference-time energy consumption in 3D-LLMs.

**Gradient conflict optimization.** When jointly optimizing $\mathcal{L}_{\text{dispersion}}$ and $\mathcal{L}_{\text{persistence}}$, their gradients may conflict. To stabilize the optimization, we adopt a projection-based adjustment inspired by PCGrad (Yu et al., 2020). Given gradients $g_{\text{disp}}$ and $g_{\text{pers}}$, if their inner product is negative, we remove the conflicting component:

$$g_{\text{disp}} \leftarrow g_{\text{disp}} - \frac{\langle g_{\text{disp}}, g_{\text{pers}} \rangle}{\|g_{\text{pers}}\|^2 + \epsilon}\, g_{\text{pers}}. \tag{11}$$

The final update direction is then

$$g_{\text{final}} = g_{\text{disp}} + g_{\text{pers}}. \tag{12}$$

This adjustment mitigates destructive interference and improves optimization stability.

## 4 EXPERIMENTS

### 4.1 EXPERIMENTAL CONFIGURATIONS

**Datasets and Models.** We conduct experiments on two point-cloud datasets: Objaverse (Deitke et al., 2023), a large-scale collection of diverse 3D objects with rich annotations, and Model-Net40 (Wu et al., 2015), a widely used benchmark of CAD-based 3D models. For each dataset, we uniformly sample 8,192 points per object and randomly select 500 point clouds as the evaluation set. For 3D-LLMs, we consider four representative models that take raw point clouds as input: PointLLM (Xu et al., 2024), X-InstructBLIP (Panagopoulou et al., 2023), GreenPLM (Tang et al., 2025), and MiniGPT-3D (Tang et al., 2024), ensuring that the evaluation focuses purely on point-cloud understanding without interference from auxiliary modalities.

**Comparison Baselines.** To the best of our knowledge, we are the first to investigate resource exhaustion attacks against 3D-LLMs. Since no prior work explicitly targets resource-oriented degradation, we compare Exhaust3D against Gaussian Noise, which adds random perturbations to point coordinates, and Random Drop, which removes 10% of points. These baselines simulate realistic point cloud corruptions commonly encountered in practice and generally affect 3D understanding. Comparing with them highlights that our method specifically targets inference-time computation rather than recognition accuracy.

Table 1: Resource overhead of four 3D-LLMs on Objaverse and ModelNet40 under different attack settings. Best results are in **bold**.

| Model | Method | Objaverse | | | ModelNet40 | | |
|---|---|---|---|---|---|---|---|
| | | Length | Latency | Energy | Length | Latency | Energy |
| PointLLM | Original | 19.77 | 0.84 | 66.45 | 14.76 | 0.65 | 52.45 |
| | Gaussian Noise | 20.58 | 1.08 | 104.78 | 22.84 | 0.89 | 70.62 |
| | Random Drop | 25.27 | 1.32 | 127.95 | 19.26 | 0.78 | 62.32 |
| | Exhaust3D | **127.52** | **5.65** | **406.50** | **45.90** | **1.88** | **139.27** |
| X-InstructBLIP | Original | 16.87 | 1.76 | 98.65 | 10.04 | 1.62 | 88.52 |
| | Gaussian Noise | 11.86 | 1.22 | 72.68 | 7.91 | 1.67 | 92.34 |
| | Random Drop | 16.62 | 1.72 | 100.35 | 10.12 | 1.62 | 89.29 |
| | Exhaust3D | **39.03** | **2.83** | **272.43** | **32.85** | **2.40** | **193.54** |
| MiniGPT-3D | Original | 27.84 | 6.92 | 183.96 | 11.34 | 6.93 | 239.10 |
| | Gaussian Noise | 12.98 | 6.91 | 181.70 | 13.00 | 6.92 | 243.54 |
| | Random Drop | 26.28 | 6.87 | 190.14 | 11.34 | 6.92 | 237.48 |
| | Exhaust3D | **74.71** | **6.79** | **553.37** | **65.87** | **6.71** | **383.17** |
| GreenPLM | Original | 16.52 | 0.92 | 43.03 | 13.69 | 0.82 | 14.01 |
| | Gaussian Noise | 16.25 | 1.04 | 45.84 | 16.69 | 0.91 | 15.26 |
| | Random Drop | 17.67 | 1.09 | 47.80 | 13.93 | 0.84 | 14.46 |
| | Exhaust3D | **41.48** | **1.52** | **67.10** | **35.58** | **1.50** | **59.82** |

**Evaluation Metrics.**    Our evaluation considers efficiency-oriented metrics, including average output sequence length, inference latency (s), and energy consumption (J), which directly reflect the ability of an attack to exhaust computational resources.

**Implementation Details.**    For all models, the default prompts are applied, and the maximum output length is set to 2,048 tokens. Exhaust3D is optimized for 100 iterations with a perturbation bound of $\ell_\infty \leq 0.1$. Gaussian Noise is also applied with the same $\ell_\infty \leq 0.1$ bound, while Random Drop directly removes 10% of the points. Additional architectural settings and implementation details are provided in Appendix A.

### 4.2 MAIN RESULTS

Table. 1 evaluates the resource overhead induced by different perturbations on four 3D-LLMs across Objaverse and ModelNet40. Exhaust3D consistently produces the most severe increases, while Gaussian Noise and Random Drop have limited effects.

On **Objaverse**, Exhaust3D significantly inflates PointLLM's response length (from 20 to 128 tokens, 6.5×), latency (6.7×), and energy (512% ↑). MiniGPT-3D also shows notable overhead, whereas X-InstructBLIP exhibits moderate growth and GreenPLM remains relatively robust (costs roughly double).

On **ModelNet40**, the same pattern holds with slightly smaller magnitudes: PointLLM and MiniGPT-3D are most affected, X-InstructBLIP sees moderate inflation, and GreenPLM is stable.

Overall, PointLLM and MiniGPT-3D are consistently the most vulnerable to Exhaust3D, X-InstructBLIP shows moderate sensitivity, and GreenPLM demonstrates strong resilience, highlighting efficiency robustness as a key consideration for 3D-LLMs.

To further illustrate these effects, we analyze the distribution of output length, latency, and energy across all perturbations(Figure. 3. Interestingly, all three metrics under Exhaust3D exhibit a bimodal distribution: one peak remains close to the baseline responses of the original inputs, while another peak emerges at significantly higher values. This pattern suggests that the attack not only shifts the average cost upward but also creates highly variable responses, with some inputs remaining

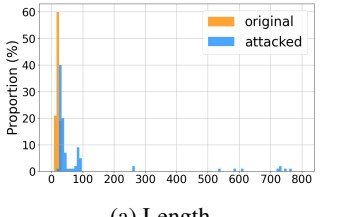 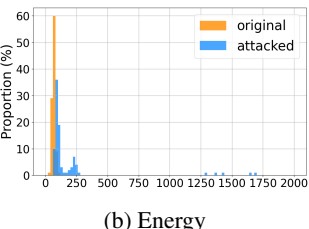 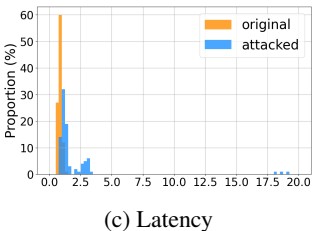

(a) Length        (b) Energy        (c) Latency

Figure 3: Distribution of PointLLM results on Objaverse: (a) Sequence Length, (b) Energy Consumption, and (c) Inference Latency.

relatively stable while others trigger extreme resource escalation. Such bimodality highlights the instability of 3D-LLMs under targeted perturbations and the difficulty of predicting computational demands in adversarial settings.

We further conduct t-SNE analyses of the learned embeddings to gain an interpretable perspective. At the **token level**, Exhaust3D induces clear structural changes in the feature space: clusters become more concentrated and compact compared to the dispersed structure in the original setting, reflecting that token semantics are significantly altered by the perturbation. At the **instance level**, however, the global layout of embeddings remains relatively close to the original distribution, indicating that the overall point cloud appearance is preserved. Taken together, these results reveal that Exhaust3D achieves a favorable trade-off: it maintains a degree of visual imperceptibility at the

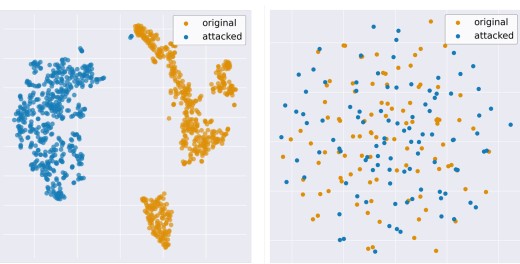

Figure 4: T-SNE visualization of learned embeddings under Exhaust3D. Left: token-level embeddings become more compact. Right: instance-level embeddings retain the overall layout.

instance level while substantially disrupting token-level semantic alignment, thereby driving the observed escalation in sequence length, latency, and energy.

Table 2: Ablation study of loss objectives for PointLLM on the Objaverse dataset. Best results are in **bold**.

| $\mathcal{L}_{\text{dispersion}}$ | $\mathcal{L}_{\text{persistence}}$ | PCGrad | Length | Latency | Energy |
|:---:|:---:|:---:|:---:|:---:|:---:|
| ✓ | | | 101.85 | 4.62 | 320.79 |
| | ✓ | | 31.42 | 1.53 | 116.79 |
| ✓ | ✓ | | 124.25 | 5.21 | 336.17 |
| ✓ | ✓ | ✓ | **127.52** | **5.65** | **406.50** |

### 4.3 ABLATION STUDY

**Effect of Loss Objectives.** We first investigate the contribution of each loss objective in Exhaust3D. Table 2 reports the results for $\mathcal{L}$dispersion, $\mathcal{L}$persistence, and PCGrad, individually and in combination. Both $\mathcal{L}$dispersion and $\mathcal{L}$persistence independently improve attack effectiveness, increasing sequence length and energy consumption, with $\mathcal{L}_{\text{dispersion}}$ having a more decisive impact due to its substantially higher numerical gains. Combining the two further enhances performance, while the addition of PCGrad yields the most pronounced effect, achieving the longest outputs, highest latency, and largest energy overhead. This confirms that all three components contribute positively and synergistically to Exhaust3D's efficiency attack.

**Effect of Masking Strategy.** We next assess the importance of the **semantic-aware point mask** by comparing it with a random mask of identical sparsity (Table 3). While random masking can still produce moderate perturbation effects, our semantic-aware mask consistently focuses on points associated with top-ranked latent tokens, yielding slightly stronger attack effects. Visual inspection

Table 3: Ablation study on the mask method.

| Method | Length | Latency | Energy |
|--------|--------|---------|--------|
| Original | 14.76 | 0.65 | 52.45 |
| Random Mask | 42.85 | 1.88 | 139.27 |
| Ours | 45.90 | 2.25 | 148.46 |

Table 4: Ablation study on the mask ratio.

| Mask ratio | Length | Energy | Latency | $l_2$ |
|------------|--------|--------|---------|-------|
| 30% | 32.68 | 1.39 | 108.52 | 0.38 |
| 50% | 45.90 | 1.88 | 139.27 | 0.45 |
| 100% | 44.06 | 1.79 | 141.61 | 0.70 |

Table 5: Ablation study on the effect of point number.

| Points | Objaverse | | | ModelNet40 | | |
|--------|--------|--------|---------|--------|--------|---------|
| | Length | Energy | Latency | Length | Energy | Latency |
| 2048 | 52.10 | 2.15 | 154.04 | 46.38 | 2.44 | 179.23 |
| 4096 | 140.71 | 7.71 | 503.01 | 45.91 | 1.88 | 138.54 |
| 8192 | 127.52 | 5.65 | 406.50 | 45.90 | 1.88 | 139.27 |

further confirms that the mask tends to cover informative regions, enhancing the attack impact while maintaining imperceptibility (see Appendix B for detailed visualization).

**Effect of Mask Ratio.** We conduct an ablation study on the mask ratio to examine its effect on attack behavior (Table 4). The results reveal a clear trade-off: smaller mask ratios yield more imperceptible perturbations with reduced $\ell_2$ norms, while moderate ratios enhance attack effectiveness with only a slight loss in stealthiness. However, overly large ratios do not bring proportional gains and instead increase perturbation magnitude, suggesting that balanced sparsity is key to achieving both efficiency and imperceptibility.

**Effect of Point Cloud Size.** We further study the impact of point count on Exhaust3D (Table 5). On Objaverse, the attack weakens significantly at 2,048 points, with much shorter responses and lower overhead compared to denser inputs. This is likely because Objaverse objects contain fine-grained structures that become underrepresented at lower resolutions, limiting the effectiveness of semantic-aware perturbations. In contrast, ModelNet40 exhibits relatively stable results across all point counts. Since its CAD-style objects are already coarse and lack detailed geometry, reducing the number of points does not substantially change the attack outcome.

**Summary.** Overall, these ablation studies demonstrate that (i) the combination of loss objectives with PCGrad maximizes attack efficiency, (ii) semantic-aware masking plays a key role in selecting high-impact points while preserving imperceptibility, (iii) smaller mask ratios yield more stealthy perturbations without fully sacrificing attack strength, and (iv) the effect of point count depends on dataset granularity: attacks weaken on Objaverse at low resolutions due to loss of fine-grained details, whereas ModelNet40 remains stable given its inherently coarse geometry. These findings highlight the design choices that underpin Exhaust3D's effectiveness and adaptability across diverse scenarios.

## 5 CONCLUSION

In this work, we presented **Exhaust3D**, the first adversarial framework that performs semantic-aware resource exhaustion attacks against 3D-LLMs through purely 3D-modal perturbations. By integrating semantic-aware adversarial manipulation with a trajectory disruption mechanism, Exhaust3D induces verbose decoding and substantially escalates inference-time energy consumption while remaining imperceptible to human observers. Our experiments across multiple datasets, point cloud resolutions, and masking strategies confirm both the effectiveness and generalizability of the method, exposing a critical vulnerability in 3D multimodal reasoning systems and underscoring the need for future research on efficiency-aware defense mechanisms.

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

APPENDIX

## A    MORE EXPERIMENTAL DETAILS

In this work, we focus on **3D point clouds** as the input modality (instead of projecting them into images or multi-view representations). Accordingly, we adopt several open-source and competitive 3D-to-text models as baselines. Below we describe their architectural details and the common implementation protocols.

**PointLLM Settings**    The adopted PointLLM (Xu et al., 2024) is composed of a Point-BERT (Yu et al., 2022) encoder pre-trained with ULIP-2 (Xue et al., 2024), a multi-layer projector with GeLU activations to map point features into the embedding space, and a Vicuna-7B (Chiang et al., 2023) LLM as the language backbone. With two additional special tokens, the vocabulary size is 32003. Prompt template used in PointLLM is ``Describe the 3D model.''.

**X-InstructBLIP Settings**    The adopted X-InstructBLIP (Panagopoulou et al., 2023) integrates a Point-BERT (Yu et al., 2022) encoder pre-trained with ULIP-2 (Xue et al., 2024) for 3D representation learning, a Q-Former module that learns query tokens, and a Vicuna-7B (Chiang et al., 2023) as the language backbone. All Q-Formers are initialized with BLIP-2 (Li et al., 2023b) stage-1 weights to ensure stable training and effective multimodal alignment. Prompt template used in X-InstructBLIP is ``Describe the 3D model.''.

**MiniGPT-3D Settings**    The adopted MiniGPT-3D (Tang et al., 2024) is composed of a Q-Former initialized from BLIP-2 (Li et al., 2023b), a Mixture of Query Experts (MQE) to enhance semantic representation, a modality projector for aligning point queries with the text embedding space, and the Phi-2-2.7B backbone (Javaheripi et al., 2023) as the language model. Prompt template used in MiniGPT-3D is ``Describe the 3D model in short''.

**GreenPLM Settings**    The adopted GreenPLM (Tang et al., 2025) consists of a ViT (Dosovitskiy et al., 2020) point encoder and an EVA-CLIP-E (Sun et al., 2023) text encoder, both trained by Uni3D (Zhou et al., 2023), a two-layer MLP projector with GeLU activation to align encoder outputs with the language embedding space, and the Phi-3 (Abdin et al., 2024) model as the language backbone. Prompt template used in GreenPLM is ``Describe the 3D model in detail''.

**Implementation Notes**

- All experiments are conducted with FP16 precision.
- Each point cloud is uniformly sampled into 8192 points (except for ablation studies on point number).
- ModelNet40 objects are assigned a fixed black color to compensate for missing texture.
- All models are evaluated in their default inference mode (no fine-tuning for our attack).
- We ensure that all models receive the same point sampling (e.g., fixed 8,192 points unless testing point-number ablations).
- For metrics like latency and energy, we run each model under the same hardware setting (single H20 GPU) and average over multiple runs.

## B    VISUALIZATION OF OUR SAMANTIC-AWARE MASK

To qualitatively illustrate the behavior of our semantic-aware mask, we visualize its effect on representative point clouds from both Objaverse (Deitke et al., 2023) (Figure 5) and ModelNet40 (Wu et al., 2015) (Figure 6). In each figure, the top row shows the point cloud with the semantic-aware mask applied, highlighting the points selected for perturbation, while the bottom row shows the original, unmasked point cloud for reference.

As can be observed, the semantic-aware mask consistently covers the core regions of the objects that are strongly associated with their semantic meaning. For example, in the Objaverse samples

(Figure 5), the mask predominantly highlights the body and characteristic parts of objects, leaving peripheral or less informative points unaltered. Similarly, in ModelNet40 (Figure 6), masked points focus on the geometrically and semantically salient regions, such as the roof and legs of chairs or the wings and fuselage of airplanes. This visualization confirms that our mask selectively targets points that are most influential for the model's understanding, ensuring that perturbations concentrate on semantically critical areas while minimally affecting irrelevant points.

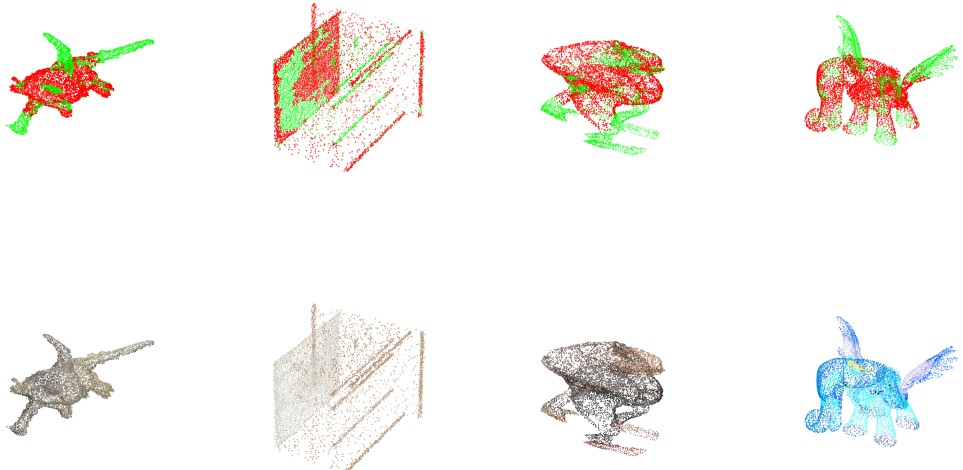

Figure 5: The visualization of our semantic-aware mask on Objaverse. The top row shows the masked object, while the bottom row shows the original object without mask. In the masked objects, points covered by the semantic-aware mask are highlighted in red, while the remaining unmasked points are shown in green.

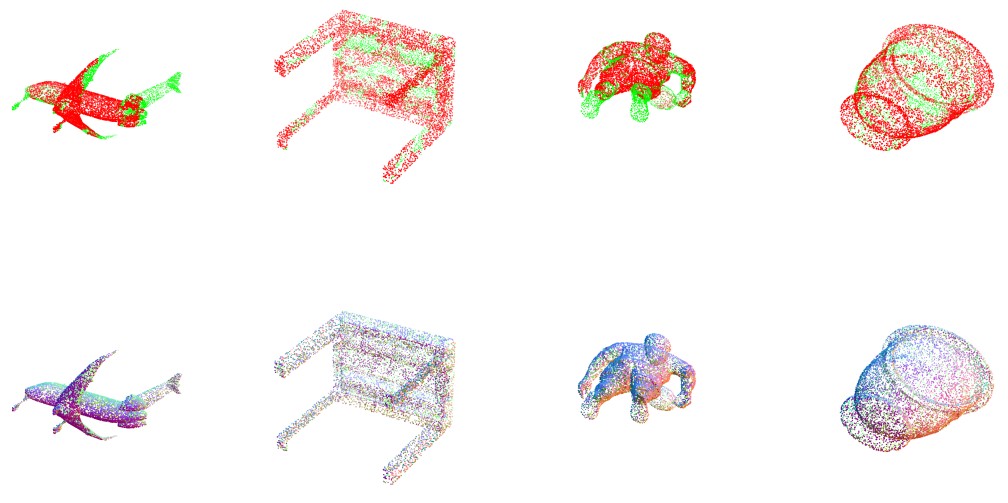

Figure 6: The visualization of our semantic-aware mask on ModelNet40. The top row shows the masked object, while the bottom row shows the original object without mask. In the masked objects, points covered by the semantic-aware mask are highlighted in red, while the remaining unmasked points are shown in green.

## C   LLM USAGE

We used an OpenAI LLM (GPT-5) as a writing and formatting assistant. In particular, it helped refine grammar and phrasing, improve clarity, and suggest edits to figure/table captions and layout (e.g., column alignment, caption length, placement). The LLM did not contribute to research ideation, experimental design, implementation, data analysis, or technical content beyond surface-level edits. All outputs were reviewed and edited by the authors, who take full responsibility for the final text and visuals.

