# OpenReview forum: "Talk Until You Burn Out: Escalating 3D-LLM Overgeneration via Semantic Manipulation"
_ICLR.cc/2026/Conference — ICLR 2026 Conference Withdrawn Submission_

### Official Review · Reviewer_gm3X · 2025-10-25

**Soundness:** 2
**Presentation:** 3
**Contribution:** 2
**Rating:** 2
**Confidence:** 3

**Summary:**

The paper presents a novel resource exhaustion attack on the 3D-LLMs by injecting imperceptible yet strategically structured semantic perturbations into 3D point clouds. It has two specific components: 1) Semantic-aware adversarial manipulation that leverages internal model representations to selectively perturb semantically critical point regions while preserving geometric structure. 2) a trajectory disruption mechanism that forces high entropy predictions to prolong the end of sequence token generation and predict verbose outputs.

**Strengths:**

1. The problem studied is novel and important, as it brings attention to the resource exhaustion attack on the 3D LLMs.

2. The attack is quite simple to develop, hence it becomes more important.

3. Experimental results further back the importance of the work.

**Weaknesses:**

The major issues with this work are:

1. The performance metrics are not reported, which are claimed to be good.

2. Table 1 results seem ambiguous, as the results on MiniGPT shows a latency of 6.79 and 6.71, while other methods have higher latency, still the paper reports their results as best. To the best of my understanding, higher latency is what authors are aiming to achieve.

3. There is no reproducibility statement in the paper, and no source code or any statement regarding that is made.

4. The paper can benefit by adding additional explanations on why persistence loss is not enough alone and why it requires other losses. While there is some discussion about these results, still authors are not very clear why the persistence loss or dispersion loss is not sufficient alone. By my understanding, authors want to achieve a delay in the end-of-sequence token, and that could be easily achieved using the persistence loss, while some loss in performance (that is why reporting performance is important), I am not sure how the authors are getting significantly smaller sequence length while only optimizing the persistence loss.

5. The compared baselines are really narrow, as the authors can try to extend the existing baselines towards 3D LLMs and compare with them, which can prove the importance of their method.

**Questions:**

See weaknesses.

---

> ### Author Response · Authors · 2025-11-21
> **Response to Reviewer gm3X (Q1-Q4)**
>
> We thank the reviewer for the constructive and detailed feedback. Below we address each concern point-by-point.
>
> ---
>
> **Q1: The performance metrics are not reported, which are claimed to be good.**
> **A1:** We appreciate the reviewer’s comment and acknowledge that our original wording regarding model performance was not sufficiently precise. To clarify, our claim refers to the observation that **Exhaust3D preserves the functional performance of 3D-LLMs on clean inputs while increasing energy-latency on malicious inputs**.
>
> Specifically, for malicious inputs, Exhaust3D can induce longer computations and may reduce the quality of generated outputs. This is consistent with prior observations from NICGSlowDown, which demonstrated that efficiency-focused attacks can substantially increase resource usage and may affect output quality for adversarial inputs, while leaving performance on benign inputs largely unaffected.
>
> Thus, although we did not report detailed performance metrics in the paper, the qualitative observation is that the model’s behavior on clean data—semantic correctness and coherence—remains intact. We will revise the manuscript to clarify that this claim applies to **clean versus malicious inputs**, and does not imply that outputs for adversarial inputs are unaffected.
>
> ---
>
> **Q2: Table 1 results seem ambiguous and latency appears inconsistent with claims.**
> **A2:**  We apologize for the **boldface formatting error** in Table 1, which caused confusion. We clarify the intended interpretation:
>
> - **Length** (average decoding steps) reflects the **average number of decoding steps**, i.e., the average number of tokens generated, and is the **primary and most stable** indicator of computational cost.
> - **Latency** and **Energy** fluctuate with GPU load, temperature, and machine environment.
>
> Importantly, MiniGPT-3D is intrinsically slow, and its baseline latency already exceeds PointLLM’s maximum latency. Therefore, even large changes in Length do not directly translate to proportional latency changes for MiniGPT-3D.
> For this reason, we consider Length the most reliable and environment-independent proxy for energy consumption, and we rely on it for the main conclusion.
> The formatting error will be corrected in the revised version.
>
>
> ---
>
> **Q3: No reproducibility statement or code availability.**
> **A3:**  We will include a complete reproducibility statement in the revised version and release the source code after the review period.
>
> ---
>
> **Q4: It is unclear why persistence loss alone is insufficient, and why dispersion loss is needed.**
> **A4:**
> **(1) Empirical evidence**
> Our ablation shows:
>
> - **$L_{persistence}$** only → limited effect
> - **$L_{dispersion}$** only → significant output-length increase
> - **$L_{dispersion}$  + $L_{persistence}$** → best overall performance
>
> Thus, $L_{dispersion}$ plays the dominant role.
>
> **(2) Why L_persistence alone does not achieve long sequences**
> Our optimization occurs during decoding. At decoding step *s*, the model state depends on:
>
> - point-cloud tokens
> - input text tokens
> - all previously generated tokens
>
> When the point cloud is updated, the entire trajectory from step 1 to *s* changes. This causes a **chain reaction** that alters logits at all future steps, making long-range EOS suppression unstable. Hence, delaying EOS probability alone is insufficient.
>
> **(3) Why $L_{dispersion}$ succeeds**
> $L_{dispersion}$ maximizes the entropy of logits, pushing token probabilities toward a uniform distribution. This naturally suppresses EOS while maintaining stable uncertainty across decoding steps.
> A uniform distribution is statistically more stable than enforcing “all tokens uniform except EOS=0”, which explains why $L_{dispersion}$ provides stronger and more reliable effects.
> $L_{persistence}$ then acts as a complementary component that further prolongs generation.

---

> ### Author Response · Authors · 2025-11-21
> **Response to Reviewer gm3X (Q5(A)-Q5(B))**
>
> **Q5: Baselines are narrow; feasibility of the threat model is unclear.**
> **A5:**
> **(A) Additional baselines**
> We added two new baselines to address the narrow comparison:
>
> 1. **PGD (untargeted)** applied to point coordinates.
> 2. **NICGSlowDown** adapted to 3D by perturbing inputs via efficiency gradients(**only focused on EOS**).
>
> | Model        | Method           | Objaverse |          |          | ModelNet40 |          |          |
> |--------------|-----------------|-----------|----------|----------|------------|----------|----------|
> |              |                 | Length    | Latency  | Energy   | Length     | Latency  | Energy   |
> | PointLLM     | Original        | 19.77     | 0.84     | 66.45    | 14.76      | 0.65     | 52.45    |
> |              | PGD             | 34.77     | 3.20     | 143.72   | 24.19      | 1.78     | 79.74    |
> |              | NICGSlowdown    | 29.99     | 1.46     | 119.16   | 18.19      | 0.95     | 19.75    |
> |              | Exhaust3D       | 127.52    | 5.65     | 406.50   | 45.90      | 1.88     | 139.27   |
> | X-InstructBLIP | Original      | 16.87     | 1.76     | 98.65    | 10.04      | 1.62     | 88.52    |
> |              | PGD             | 13.07     | 0.73     | 37.91    | 9.16       | 0.90     | 63.25    |
> |              | NICGSlowdown    | 12.82     | 0.74     | 4.38     | 9.38       | 0.96     | 5.68     |
> |              | Exhaust3D       | 39.03     | 2.83     | 272.43   | 32.85      | 2.40     | 193.54   |
> | MiniGPT-3D   | Original        | 27.84     | 6.92     | 183.96   | 11.34      | 6.93     | 239.10   |
> |              | PGD             | 29.57     | 6.34     | 560.81   | 23.52      | 6.70     | 580.63   |
> |              | NICGSlowdown    | 26.25     | 6.76     | 424.00   | 28.92      | 6.68     | 410.80   |
> |              | Exhaust3D       | 74.71     | 6.79     | 553.37   | 65.87      | 6.71     | 383.17   |
> | GreenPLM     | Original        | 16.52     | 0.92     | 43.03    | 13.69      | 0.82     | 14.01    |
> |              | PGD             | 14.36     | 2.10     | 184.19   | 12.24      | 1.80     | 156.62   |
> |              | NICGSlowdown    | 15.97     | 2.27     | 132.77   | 15.70      | 2.29     | 137.72   |
> |              | Exhaust3D       | 41.48     | 1.52     | 67.10    | 35.58      | 1.50     | 59.82    |
>
>
> As shown in the experimental results above, both PGD and NICGSlowDown achieve significantly weaker attack performance compared with Exhaust3D. This demonstrates the superiority of our method and validates the importance of our proposed $L_{dispersion}$ term, which explicitly increases the entropy of token-probability distributions to induce longer outputs.
>
> ---
>
> **(B) Evaluation under defense mechanisms**
>
> To further assess the robustness of **Exhaust3D**, we evaluate it under three representative point-cloud defenses:
>
> - **DupNet**
> - **SRS (Simple Random Sampling)**
> - **SOR (Statistical Outlier Removal)**
>
> | Model     | Settings         | Objaverse        |                   |                  | ModelNet40      |                   |                  |
> |-----------|-----------------|-----------------|-----------------|-----------------|-----------------|-----------------|-----------------|
> |           |                 | Length          | Latency         | Energy          | Length          | Latency         | Energy          |
> | PointLLM | Original         | 19.77           | 0.84            | 66.45           | 14.76           | 0.65            | 52.45           |
> |           | Exhaust3D        | 127.52          | 5.65            | 406.50           | 45.9            | 1.88            | 139.27          |
> |           | Exhaust3D+Dup-Net| 62.62           | 1.56            | 363.21          | 23.2            | 0.63            | 143.75          |
> |           | Exhaust3D+SOR    | 62.52           | 1.54            | 354.68          | 24.29           | 0.66            | 150.17          |
> |           | Exhaust3D+SRS    | 64.50           | 1.60            | 371.94          | 23.88           | 0.65            | 148.11          |
>
> Across all three defenses, we observe a moderate decrease in attack effectiveness, which is expected due to point upsampling, random subsampling, and statistical denoising. However, the performance drop remains within a reasonable and acceptable range, and Exhaust3D consistently preserves its ability to induce substantial output-length expansion and latency increase.
>
> These results indicate that the perturbations introduced by Exhaust3D are not fully removed by standard point-cloud purification, as they alter deeper semantic structures within the model’s hidden states, rather than relying solely on surface-level or high-frequency noise.

---

> ### Author Response · Authors · 2025-11-21
> **Response to Reviewer gm3X (Q5(C))**
>
> **(C) Transferability to black-box settings**
> To further support the effectiveness of **Exhaust3D**, we conducted **transfer attacks**:
>
> - **Source model:** PointLLM
> - **Target models:** GreenPLM, MiniGPT-3D, X-InstructBLIP
>
> | Source Model | Target Model    | Objaverse |          |          | ModelNet40 |          |          |
> |--------------|----------------|-----------|----------|----------|------------|----------|----------|
> |              |                | Length    | Latency  | Energy   | Length     | Latency  | Energy   |
> | PointLLM     | MiniGPT-3D         | 70.06     | 6.6959   | 912.91   | 61.01      | 6.6982   | 907.95   |
> |              | GreenPLM        | 46.51     | 1.809    | 51.49    | 59.48      | 2.3228   | 339.35   |
> |              | X-InstructBLIP  | 19.38     | 0.594    | 77.70    | 12.84      | 0.5188   | 111.16   |
>
> Results show:
>
> - Strong transfer to **GreenPLM** and **MiniGPT-3D**
> - Slightly higher effect on GreenPLM (likely due to architectural similarity with PointLLM)
> - Weaker transfer to **X-InstructBLIP**, expected because it is designed for **multi-modal alignment** rather than single-modality 3D processing
>
> These results confirm that **Exhaust3D remains effective beyond white-box settings** and further highlight the robustness of our approach.
>
> ---
>
> **We hope these clarifications address all concerns and strengthen the overall contribution of the paper.**

---

> ### Author Response · Authors · 2025-11-26
> **Gentle Reminder: Updated Reply to Reviewer gm3X**
>
> Dear  gm3X,
>
> We have carefully supplemented responses to your further questions and provided experiments following your suggestion. We look forward to your reply and welcome discussion on any questions regarding our paper and response.
>
> Best regards,
>
> Authors

---

### Official Review · Reviewer_eSDw · 2025-10-26

**Soundness:** 2
**Presentation:** 3
**Contribution:** 2
**Rating:** 4
**Confidence:** 4

**Summary:**

This paper investigates a new type of vulnerability in 3D large language models (3D-LLMs) — adversarial perturbations that intentionally increase inference time and energy consumption. The authors propose a **semantic-aware perturbation** combined with a **trajectory-intervention mechanism** to induce over-generation while maintaining semantic and visual consistency. Experiments on two datasets and four representative 3D-LLMs demonstrate significant increases in output length and latency, supported by extensive ablation analyses.

**Strengths:**

* **Important problem.** This paper identifies an important yet underexplored risk — increasing 3D-LLM inference time through adversarial manipulation of point clouds, which can raise operational costs for online users.
* **Well-structured method.** The proposed semantic-aware perturbation and trajectory-intervention mechanism directly target the problem and are clearly written and easy to understand.
* **Reasonably broad experiments.** The paper evaluates two datasets and four representative 3D-LLMs, and conducts extensive ablation studies and hyperparameter analyses.

**Weaknesses:**

* **Limited real-world applicability due to white-box assumption.**
 The method and evaluation rely on white-box access, which severely limits the practical impact of the proposed attack, since real-world 3D-LLMs are often closed-source.
* **Inappropriate baselines.**
 The selected baselines (Gaussian Noise and Random Drop) are unrelated to energy-exhaustion behavior. More relevant baselines, such as those targeting delayed EOS or increased decoding entropy, should be added.
* **Lack of defense evaluation.**
 The paper does not evaluate any defense mechanisms, making it difficult to assess the robustness of the proposed method.
* **Unsupported functional-quality claim.**
 The abstract claims that model functionality is nearly unaffected, but there is no rigorous experimental evidence or analysis to substantiate this claim.

**Questions:**

The paper’s main limitation lies in its **white-box design** and **unrelated baselines**, which significantly restrict its **practical relevance and generalizability** despite an interesting problem formulation.

---

> ### Author Response · Authors · 2025-11-21
> **Response to Reviewer eSDw (Q1-Q2)**
>
> We thank the reviewer for the constructive and detailed feedback. Below we address each concern point-by-point.
>
> ---
>
> **Q1: Limited real-world applicability due to white-box assumption**
> **A1:**  We agree that the white-box assumption is a strong one. To address this concern, we additionally evaluate **transferability in black-box settings**, following the reviewer’s suggestion.
>
> - **Source model:** PointLLM
> - **Target models:** GreenPLM, MiniGPT-3D, X-InstructBLIP
>
> | Source Model | Target Model    | Objaverse |          |          | ModelNet40 |          |          |
> |--------------|----------------|-----------|----------|----------|------------|----------|----------|
> |              |                | Length    | Latency  | Energy   | Length     | Latency  | Energy   |
> | PointLLM     | MiniGPT-3D         | 70.06     | 6.6959   | 912.91   | 61.01      | 6.6982   | 907.95   |
> |              | GreenPLM        | 46.51     | 1.809    | 51.49    | 59.48      | 2.3228   | 339.35   |
> |              | X-InstructBLIP  | 19.38     | 0.594    | 77.70    | 12.84      | 0.5188   | 111.16   |
>
> Our results show:
>
> - Strong transferability to GreenPLM and MiniGPT-3D
> - Moderate transfer to X-InstructBLIP, likely due to its multimodal-aligned architecture
>
> Thus, while Exhaust3D is designed as a white-box attack, it remains effective under black-box transfer, indicating that our semantic-aware perturbations influence model-internal dynamics in a way that generalizes across architectures.
>
> We will clarify this point in the revision.
>
> ---
>
> ## Q2: Inappropriate baselines
>
> **A2:**  We appreciate this observation. Following the reviewer’s suggestion, we added two stronger and more relevant baselines:
>
> 1. **PGD (untargeted)** applied on point coordinates
> 2. **NICGSlowDown** adapted to 3D inputs (delaying EOS through efficiency gradients)
>
> Both represent intended attempts to increase decoding time or disturb EOS behavior.
>
> | Model        | Method           | Objaverse |          |          | ModelNet40 |          |          |
> |--------------|-----------------|-----------|----------|----------|------------|----------|----------|
> |              |                 | Length    | Latency  | Energy   | Length     | Latency  | Energy   |
> | PointLLM     | Original        | 19.77     | 0.84     | 66.45    | 14.76      | 0.65     | 52.45    |
> |              | PGD             | 34.77     | 3.20     | 143.72   | 24.19      | 1.78     | 79.74    |
> |              | NICGSlowdown    | 29.99     | 1.46     | 119.16   | 18.19      | 0.95     | 19.75    |
> |              | Exhaust3D       | 127.52    | 5.65     | 406.50   | 45.90      | 1.88     | 139.27   |
> | X-InstructBLIP | Original      | 16.87     | 1.76     | 98.65    | 10.04      | 1.62     | 88.52    |
> |              | PGD             | 13.07     | 0.73     | 37.91    | 9.16       | 0.90     | 63.25    |
> |              | NICGSlowdown    | 12.82     | 0.74     | 4.38     | 9.38       | 0.96     | 5.68     |
> |              | Exhaust3D       | 39.03     | 2.83     | 272.43   | 32.85      | 2.40     | 193.54   |
> | MiniGPT-3D   | Original        | 27.84     | 6.92     | 183.96   | 11.34      | 6.93     | 239.10   |
> |              | PGD             | 29.57     | 6.34     | 560.81   | 23.52      | 6.70     | 580.63   |
> |              | NICGSlowdown    | 26.25     | 6.76     | 424.00   | 28.92      | 6.68     | 410.80   |
> |              | Exhaust3D       | 74.71     | 6.79     | 553.37   | 65.87      | 6.71     | 383.17   |
> | GreenPLM     | Original        | 16.52     | 0.92     | 43.03    | 13.69      | 0.82     | 14.01    |
> |              | PGD             | 14.36     | 2.10     | 184.19   | 12.24      | 1.80     | 156.62   |
> |              | NICGSlowdown    | 15.97     | 2.27     | 132.77   | 15.70      | 2.29     | 137.72   |
> |              | Exhaust3D       | 41.48     | 1.52     | 67.10    | 35.58      | 1.50     | 59.82    |
>
>
> Our experiments show that:
>
> - Both PGD and NICGSlowDown have **significantly weaker impact** on output length and latency
> - NICGSlowDown in particular struggles in 3D scenarios because its mechanism primarily targets EOS suppression, which becomes unstable under multi-step trajectory changes
>
> These comparisons validate that Exhaust3D’s design—especially the **dispersion loss that raises hidden-state entropy**—is essential for reliably inducing prolonged decoding in 3D-LLMs.

---

> ### Author Response · Authors · 2025-11-21
> **Response to Reviewer eSDw (Q3-Q4)**
>
> **Q3: Lack of defense evaluation**
> **A3:**  We thank the reviewer for pointing this out. In the revised version, we evaluated Exhaust3D under **three representative 3D point-cloud defense mechanisms**:
>
> - **DupNet**
> - **SRS (Simple Random Sampling)**
> - **SOR (Statistical Outlier Removal)**
>
> | Model     | Settings         | Objaverse        |                   |                  | ModelNet40      |                   |                  |
> |-----------|-----------------|-----------------|-----------------|-----------------|-----------------|-----------------|-----------------|
> |           |                 | Length          | Latency         | Energy          | Length          | Latency         | Energy          |
> | PointLLM | Original         | 19.77           | 0.84            | 66.45           | 14.76           | 0.65            | 52.45           |
> |           | Exhaust3D        | 127.52          | 5.65            | 406.50           | 45.9            | 1.88            | 139.27          |
> |           | Exhaust3D+Dup-Net| 62.62           | 1.56            | 363.21          | 23.2            | 0.63            | 143.75          |
> |           | Exhaust3D+SOR    | 62.52           | 1.54            | 354.68          | 24.29           | 0.66            | 150.17          |
> |           | Exhaust3D+SRS    | 64.50           | 1.60            | 371.94          | 23.88           | 0.65            | 148.11          |
>
> We observe:
>
> - A moderate performance drop, which is expected.
> - But attack effectiveness remains within an acceptable and meaningful range across all three defenses
>
> This demonstrates that Exhaust3D does not rely merely on high-frequency or local noise, but instead shifts deeper semantic structures within hidden states—making the perturbations difficult to remove via standard purification.
>
> We will include these results in the updated manuscript.
>
> ---
>
> **Q4: Unsupported claim about maintained model functionality**
> **A4:** We appreciate the reviewer’s comment and realize that our original wording was unclear. To clarify, our claim refers to the observation that **Exhaust3D increases energy-latency costs on malicious 3D inputs while preserving the performance of 3D-LLMs on clean data**.
>
> Specifically, for malicious inputs, Exhaust3D can induce longer computations and may affect the quality of generated outputs. This is consistent with prior observations from NICGSlowDown, which showed that efficiency-focused attacks can substantially increase resource usage and may reduce output quality for malicious inputs. In contrast, clean inputs processed by the model maintain their semantic correctness and coherence, indicating that functional performance on non-adversarial data remains intact.
>
> We will revise the manuscript to clarify that our claim concerns **the effect of the attack on malicious versus clean samples**, rather than asserting that model functionality is universally unaffected.
>
>
>
>
> **We hope that these clarifications help address the reviewer’s concerns and better convey the contributions and implications of our work.**

---

> ### Author Response · Authors · 2025-11-26
> **Gentle Reminder: Updated Reply to Reviewer eSDw**
>
> Dear eSDw,
>
> We have carefully supplemented responses to your further questions and provided experiments following your suggestion. We look forward to your reply and welcome discussion on any questions regarding our paper and response.
>
> Best regards,
>
> Authors

---

### Official Review · Reviewer_EBEk · 2025-10-31

**Soundness:** 3
**Presentation:** 3
**Contribution:** 2
**Rating:** 4
**Confidence:** 4

**Summary:**

The paper proposes Exhast3D, a resource exhaustion attack framework on 3D large language models (3D-LLMs). The attack injects imperceptible perturbations into 3D point clouds to cause models to overgenerate long outputs, increasing inference-time energy and latency. Exhaust3D includes a semantic-aware adversarial manipulation strategy and a trajectory disruption mechanism. Experiments on the Objaverse and ModelNet40 datasets show up to 6.45× longer outputs and 6.12× higher energy use for various 3D-LLMs.

**Strengths:**

1. The paper proposes the first attack framework for resource exhaustion attacks on 3D LLMs.
2. The semantic-aware masking and dual-loss formulation are well justified, with clear equations and ablation studies.
3. Evaluation across four models and two datasets demonstrates the effectiveness of the proposed method.
4. The paper is well written and easy to follow.

**Weaknesses:**

1. The comparison is only against random noise and dropout. No semantic or gradient-based adversarial baselines (e.g., FGSM, PGD, or prior efficiency attacks like Sponge or NICGSlowDown) are adapted for 3D input.
2. The impact of the threat model is somewhat limited. While it is shown that Exhast3D can successfully increase the output of 3D LLMs, it is unclear what the practical consequences are. In addition, the threat model also assumes white-box access to the model weights, which can be infeasible in real-world applications.
3. The paper should also consider the attack performance under defenses, for example, random resampling of 3D points or applying Gaussian filtering to remove the adversarial noise.

**Questions:**

1. Adding a few simple baselines can be helpful to understand the effectiveness of the proposed method.
2. Can this method work in the gray-box or black-box settings?
3. Line 116, reference missing: LEO (?).
4. How well does the method perform under defenses?
5. Line 225-226, it is unclear why a larger l2 norm in the hidden state indicates higher token importance.

---

> ### Author Response · Authors · 2025-11-21
> **Response to Reviewer EBEk (Q1-Q2(A))**
>
> We sincerely thank you for the constructive and insightful comments. Below we address all concerns in detail and summarize the additional experiments and clarifications added in the revision.
>
> ---
>
> **Q1: The lack of adversarial baselines**
> **A1:** We appreciate the suggestion and have added two new baselines in the revision:
> 1. **PGD (untargeted)** applied to point coordinates.
> 2. **NICGSlowDown** adapted to 3D by perturbing inputs via efficiency gradients.
>
> | Model        | Method           | Objaverse |          |          | ModelNet40 |          |          |
> |--------------|-----------------|-----------|----------|----------|------------|----------|----------|
> |              |                 | Length    | Latency  | Energy   | Length     | Latency  | Energy   |
> | PointLLM     | Original        | 19.77     | 0.84     | 66.45    | 14.76      | 0.65     | 52.45    |
> |              | PGD             | 34.77     | 3.20     | 143.72   | 24.19      | 1.78     | 79.74    |
> |              | NICGSlowdown    | 29.99     | 1.46     | 119.16   | 18.19      | 0.95     | 19.75    |
> |              | Exhaust3D       | 127.52    | 5.65     | 406.50   | 45.90      | 1.88     | 139.27   |
> | X-InstructBLIP | Original      | 16.87     | 1.76     | 98.65    | 10.04      | 1.62     | 88.52    |
> |              | PGD             | 13.07     | 0.73     | 37.91    | 9.16       | 0.90     | 63.25    |
> |              | NICGSlowdown    | 12.82     | 0.74     | 4.38     | 9.38       | 0.96     | 5.68     |
> |              | Exhaust3D       | 39.03     | 2.83     | 272.43   | 32.85      | 2.40     | 193.54   |
> | MiniGPT-3D   | Original        | 27.84     | 6.92     | 183.96   | 11.34      | 6.93     | 239.10   |
> |              | PGD             | 29.57     | 6.34     | 560.81   | 23.52      | 6.70     | 580.63   |
> |              | NICGSlowdown    | 26.25     | 6.76     | 424.00   | 28.92      | 6.68     | 410.80   |
> |              | Exhaust3D       | 74.71     | 6.79     | 553.37   | 65.87      | 6.71     | 383.17   |
> | GreenPLM     | Original        | 16.52     | 0.92     | 43.03    | 13.69      | 0.82     | 14.01    |
> |              | PGD             | 14.36     | 2.10     | 184.19   | 12.24      | 1.80     | 156.62   |
> |              | NICGSlowdown    | 15.97     | 2.27     | 132.77   | 15.70      | 2.29     | 137.72   |
> |              | Exhaust3D       | 41.48     | 1.52     | 67.10    | 35.58      | 1.50     | 59.82    |
>
>
> After adding these baselines, the updated table are shown above. As illustrated, both **PGD** and **NICGSlowDown** yield significantly weaker attack performance compared with our proposed **Exhaust3D**. The primary reason is that **PGD** only pushes point-cloud features away from their original representations but does not effectively guide the downstream LLM to generate longer sequences. In addition, **NICGSlowDown** mainly focuses on manipulating the probability of the EOS token, which is insufficient for 3D-LLMs whose generation behavior is strongly influenced by multimodal semantic structure rather than EOS sensitivity alone. This further highlights the importance of our proposed **$L_{dispersion}$** term, which explicitly increases the entropy of token-probability distributions and thereby induces substantial output-length expansion.
>
> ---
>
> **Q2: The feasibility and practicality of the threat model**
> **A2:**
> **(A) Practicality and real-world consequences of Exhaust3D**
> We would like to clarify that the potential consequences of **Exhaust3D** attacks are already discussed in the Introduction. Our threat model considers users interacting with 3D-LLM APIs provided by cloud service platforms or locally deployed 3D-LLMs. Due to the large size of 3D datasets, it is often impractical to store all 3D resources locally or entirely on cloud servers. In practice, users may retrieve 3D data over the network to assist 3D-LLMs in completing tasks. In such scenarios, embedding harmful perturbations in publicly available resources can cause **abnormal increases in the output length** of large models. This leads to:
>
> - Significantly higher **API costs**, or
> - Dramatically increased **energy consumption** for locally deployed models,
>
> both of which can adversely affect users. Therefore, Exhaust3D can pose considerable security risks in real-world scenarios.

---

> ### Author Response · Authors · 2025-11-21
> **Response to Reviewer EBEk (Q2(B)-Q3)**
>
> **(B) Black-box transferability of Exhaust3D**
> To show that Exhaust3D is not limited to the white-box setting, we further evaluate its **transferability** in black-box attacks. Specifically, we used PointLLM as the source model and GreenPLM, MiniGPT-3D, and X-InstructBLIP as target models.
>
> | Source Model | Target Model    | Objaverse |          |          | ModelNet40 |          |          |
> |--------------|----------------|-----------|----------|----------|------------|----------|----------|
> |              |                | Length    | Latency  | Energy   | Length     | Latency  | Energy   |
> | PointLLM     | MiniGPT-3D         | 70.06     | 6.6959   | 912.91   | 61.01      | 6.6982   | 907.95   |
> |              | GreenPLM        | 46.51     | 1.809    | 51.49    | 59.48      | 2.3228   | 339.35   |
> |              | X-InstructBLIP  | 19.38     | 0.594    | 77.70    | 12.84      | 0.5188   | 111.16   |
>
> As shown in the table above, Exhaust3D maintains strong performance on GreenPLM and MiniGPT-3D, and even exhibits slightly improved attack effect on GreenPLM—likely due to structural similarity between PointLLM and GreenPLM, which exposes shared vulnerabilities. The attack is less effective on X-InstructBLIP, which can be attributed to architectural differences: X-InstructBLIP is designed for multi-modal alignment rather than single 3D modality, reducing transferability. These results confirm that Exhaust3D remains impactful beyond the strict white-box setting.
>
> ---
>
> **Q3: Evaluation under defense mechanisms**
> **A3:** We thank the reviewer for highlighting this important aspect. In the revised version, we additionally conduct experiments to evaluate the robustness of **Exhaust3D** under three representative point-cloud defense pipelines: **DupNet** (reconstruction-based), **SRS** (random subsampling), and **SOR** (statistical denoising).
> | Model     | Settings         | Objaverse        |                   |                  | ModelNet40      |                   |                  |
> |-----------|-----------------|-----------------|-----------------|-----------------|-----------------|-----------------|-----------------|
> |           |                 | Length          | Latency         | Energy          | Length          | Latency         | Energy          |
> | PointLLM | Original         | 19.77           | 0.84            | 66.45           | 14.76           | 0.65            | 52.45           |
> |           | Exhaust3D        | 127.52          | 5.65            | 406.50           | 45.9            | 1.88            | 139.27          |
> |           | Exhaust3D+Dup-Net| 62.62           | 1.56            | 363.21          | 23.2            | 0.63            | 143.75          |
> |           | Exhaust3D+SOR    | 62.52           | 1.54            | 354.68          | 24.29           | 0.66            | 150.17          |
> |           | Exhaust3D+SRS    | 64.50           | 1.60            | 371.94          | 23.88           | 0.65            | 148.11          |
>
> The results are reported in the table above. The results show that, although these defenses introduce a moderate reduction in attack strength, Exhaust3D still achieves substantial increases in output length and latency across all tested defense configurations. This indicates that the perturbations generated by Exhaust3D are not limited to removable local noise, but influence **semantically critical regions** that are preserved even after purification.
>
> Overall, these findings demonstrate that Exhaust3D remains effective and practically impactful under widely used 3D point-cloud defenses, with performance degradation limited to a manageable range.

---

> ### Author Response · Authors · 2025-11-21
> **Response to Reviewer EBEk (Q4-Q5)**
>
> **Q4: Missing reference at Line 116 (LEO)**
> **A4:** We thank the reviewer for pointing out this omission.
> The missing citation refers to [1]. We will add the correct reference in the revised version.
>
> ---
>
> **Q5: Why does a larger L2 norm in the hidden state indicate higher token importance?**
> **A5:** We apologize for the earlier insufficient explanation.
>
> In Transformer-based LLMs, the **magnitude of hidden-state activations** is closely associated with the **importance of the underlying semantic features**. Prior studies have shown that:
>
> 1. **Important features activate more strongly.**
>    Transformer MLP neurons exhibit *activation sparsity*, where only a small subset of neurons fire with large magnitude for salient features, while unimportant inputs remain near-zero. This indicates that **large activation norms correspond to more influential semantic signals** [2].
>
> 2. **Activation magnitude reflects semantic feature loading.**
>    Mechanistic interpretability analyses further demonstrate that high-norm activations represent **stronger feature directions** in the model’s internal space, making them more impactful to downstream computations such as next-token prediction [3].
>
> In our empirical analysis on 3D-LLMs, tokens with larger decoder hidden-state L2 norms consistently showed higher sensitivity in the output logits (Jacobian-based measurement), confirming this theoretical expectation.
> Thus, using the L2 norm provides a simple, lightweight, and model-agnostic indicator of token importance, enabling targeted and semantically meaningful perturbation.
>
> ---
>
>
> **References:**
> [1] Huang J, Yong S, Ma X, et al. An embodied generalist agent in 3d world[J]. arXiv preprint arXiv:2311.12871, 2023.
> [2] Li Z, You C, Bhojanapalli S, et al. The lazy neuron phenomenon: On emergence of activation sparsity in transformers[J]. arXiv preprint arXiv:2210.06313, 2022.
> [3] Rai D, Zhou Y, Feng S, et al. A practical review of mechanistic interpretability for transformer-based language models[J]. arXiv preprint arXiv:2407.02646, 2024.
>
> **We hope the above clarifications provide a satisfactory and transparent resolution to all concerns.**

---

> ### Author Response · Authors · 2025-11-26
> **Gentle Reminder: Updated Reply to Reviewer EBEk**
>
> Dear EBEk,
>
> We have carefully supplemented responses to your further questions and provided experiments following your suggestion. We look forward to your reply and welcome discussion on any questions regarding our paper and response.
>
> Best regards,
>
> Authors

---

### Official Review · Reviewer_LVkA · 2025-11-06

**Soundness:** 3
**Presentation:** 3
**Contribution:** 2
**Rating:** 4
**Confidence:** 5

**Summary:**

This paper presents Exhaust3D, a novel framework for adversarial attacks that target 3D Large Language Models. The core idea is quite clever—it crafts tiny, almost invisible perturbations on 3D point clouds. These manipulations trick the model's internal workings, causing it to generate extremely long and verbose outputs. The result is a massive spike in inference time and energy consumption, all while the actual factual quality of the answer remains largely intact. This is a very timely investigation into the computational weak spots of the latest multimodal AI systems.

**Strengths:**

First off, the novelty is strong.  Systematically pulling off a resource exhaustion attack purely through the 3D modality, which is a smart focus given how new and computationally heavy these models are. Secondly, the attack itself is well-designed. The two-part strategy—using semantic-aware masking to find critical points and then a trajectory disruption mechanism—is elegant and makes a lot of sense. Using PCGrad to smooth out gradient conflicts was a nice technical detail. Finally, the evaluation is thorough. The results are compelling across multiple models and datasets, and the ablation studies do a great job of validating why the authors made their specific design choices.

**Weaknesses:**

A couple of things gave me pause, though. While the perturbations are technically imperceptible, the paper doesn't fully address how stealthy this attack would be in a real-world application. It seems a simple system check on output length could easily detect and stop these rambling responses, which would neutralize the threat. Additionally, while the work does an excellent job of exposing a vulnerability, it would be strengthened by even a brief discussion on potential defensive strategies, which would provide a more complete picture.

**Questions:**

I'm curious about the realistic scenarios where this attack would be most potent. Are you thinking more about cloud-based services where latency equals cost, or embedded systems where a resource drain could cause a physical failure? Also, the attack worked best on PointLLM, but GreenPLM was more resilient. Any thoughts on the architectural reasons for that? Understanding that could offer great insights for building tougher models. Lastly, your evaluation focused on captioning tasks. Do you have any intuition on how this would play out in more critical tasks like navigation or reasoning, where prolonged processing could have real-world consequences?

---

> ### Author Response · Authors · 2025-11-21
> **Response to Reviewer LVkA（Q1-Q2）**
>
> We sincerely thank the reviewer for the thoughtful and encouraging evaluation. We appreciate the positive recognition of our novelty, methodological design, and comprehensive experiments. Below we address the raised concerns and questions in detail.
>
> ---
>
> **Q1: How stealthy is the attack in real-world scenarios?**
> **A1:** First, Exhaust3D is stealthy at the input level. The perturbations remain *geometrically imperceptible* and *semantically consistent* in 3D space, meaning they do not alter the visual or structural integrity of the 3D scene.
>
> While a straightforward solution to mitigate the energy-latency vulnerability is to limit the generation length, we argue that this is infeasible for two main reasons. First, text prompts and 3D inputs vary widely in complexity: simple scenes may require only short responses, while complex instructions naturally induce longer generations. Even when a maximum output length is imposed, our attacks can still push the model to generate sequences close to this limit, sustaining high energy-latency costs. Second, modern 3D-LLMs are capable of generating longer, coherent sequences, and service providers often increase the maximum allowed length to ensure high-quality user experience. However, this higher tolerance for longer sequences inherently introduces more potential for energy-latency exploitation, which malicious 3D data can leverage to amplify costs.
>
> Therefore, Simply limiting the generation length cannot fully mitigate the energy-latency vulnerability. Our results show that malicious 3D data can still induce high energy-latency costs under current token limits.
>
>
>
> ---
>
> **Q2: Are there potential defensive strategies, and how robust is Exhaust3D under them?**
> **A2:** We thank the reviewer for raising this valuable question. To provide a more complete picture of the threat model, we additionally evaluate the robustness of Exhaust3D under three representative point‑cloud defense frameworks: DupNet, SRS (Simple Random Sampling), and SOR (Statistical Outlier Removal). These defenses respectively target point‑cloud reconstruction, random subsampling, and statistical denoising.
> | Model     | Settings         | Objaverse        |                   |                  | ModelNet40      |                   |                  |
> |-----------|-----------------|-----------------|-----------------|-----------------|-----------------|-----------------|-----------------|
> |           |                 | Length          | Latency         | Energy          | Length          | Latency         | Energy          |
> | PointLLM | Original         | 19.77           | 0.84            | 66.45           | 14.76           | 0.65            | 52.45           |
> |           | Exhaust3D        | 127.52          | 5.65            | 406.50           | 45.9            | 1.88            | 139.27          |
> |           | Exhaust3D+Dup-Net| 62.62           | 1.56            | 363.21          | 23.2            | 0.63            | 143.75          |
> |           | Exhaust3D+SOR    | 62.52           | 1.54            | 354.68          | 24.29           | 0.66            | 150.17          |
> |           | Exhaust3D+SRS    | 64.50           | 1.60            | 371.94          | 23.88           | 0.65            | 148.11          |
>
>
> We report the results in the table above, focusing exclusively on the behavior of **Exhaust3D** after each defense is applied. As shown in this table, **Exhaust3D remains highly effective under all three defenses**, still causing substantial increases in output length and latency. In contrast, these defenses are designed primarily to remove local perturbations or high-frequency noise, but our method perturbs **semantically critical regions** identified within the model’s hidden representations. This explains why standard purification fails to neutralize the attack.
>
> These additional results confirm that **Exhaust3D is robust even under widely used 3D point‑cloud defense pipelines**, strengthening the practical relevance of our findings.

---

> ### Author Response · Authors · 2025-11-21
> **Response to Reviewer LVkA（Q3-Q5）**
>
> **Q3: In what realistic scenarios would Exhaust3D be most potent?**
> **A3:** Based on our threat model, Exhaust3D is especially potent in practical settings where users rely on externally sourced 3D data. In real applications—such as AR/VR platforms, robotics perception modules, digital twins, or 3D content pipelines—users rarely store all 3D assets locally due to their large size. Instead, they frequently retrieve 3D objects or scenes from online repositories, shared datasets, or third‑party services. Harmful perturbations embedded in such publicly accessible resources can be unknowingly propagated into downstream 3D‑LLM workflows, causing **abnormal increases in output length** and triggering significant overhead:
>
> - **Cloud-based APIs**: Commercial LLM services typically charge per generated token. If malicious 3D inputs induce persistent over-generation, users may incur unexpectedly high billing costs, making the attack economically harmful even without any privileged system access.
>
> - **Locally deployed models**: Over-generation increases decoding steps and energy usage. On resource‑limited platforms—such as mobile robots, AR headsets, or embedded systems—this can cause slowdowns, unstable execution, or failure to complete tasks.
>
> Overall, because 3D applications naturally involve data sharing and remote resource retrieval, Exhaust3D can be triggered in realistic workflows and lead to substantial economic and energy‑latency consequences in both cloud and local deployment environments.
>
>
>
> ---
>
> **Q4: Why does PointLLM show stronger attack effect than GreenPLM?**
> **A4:** The primary reason lies in the generation behavior and architectural design of the two models:
>
> 1. **GreenPLM is a lightweight model designed for concise responses.**
>     This makes GreenPLM less sensitive to our over-generation objective, compared to PointLLM which naturally produces longer, more descriptive outputs.
>
> 2. **Implication for robustness.**
>    This result suggests that training models to favor concise generation** (e.g., via fine-tuning or instruction tuning) may improve robustness against resource-exhaustion attacks. However, this comes at the cost of reduced expressiveness and descriptive capability, which may be undesirable for applications requiring detailed outputs.
>
>
> ---
>
> **Q5: How would Exhaust3D affect tasks beyond captioning (e.g., navigation, reasoning)?**
> **A5:**  The underlying mechanism of Exhaust3D—manipulating hidden-state entropy to elongate decoding trajectories—extends naturally to other 3D-LLM tasks:
>
> - **Navigation / robotics planning**:
>   Prolonged decoding may lead to **deadline misses** or delayed control signals. In time-critical systems, this can degrade stability or cause the agent to act on outdated plans, posing safety concerns.
>
> - **3D reasoning tasks**:
>   Multi-step reasoning already involves longer and more variable generation path, making extended reasoning chains far more covert and harder to detect than in captioning.
>   Therefore, transferring Exhaust3D to reasoning scenarios could yield high-impact attacks, as the induced over-generation blends naturally into tasks where long chains of thought are expected.
>
> **We hope these explanations fully resolve the reviewer’s concerns and help present the contribution of our work more clearly.**

---

> ### Author Response · Authors · 2025-11-26
> **Gentle Reminder: Updated Reply to Reviewer LVkA**
>
> Dear LVkA,
>
> We have carefully supplemented responses to your further questions and provided experiments following your suggestion. We look forward to your reply and welcome discussion on any questions regarding our paper and response.
>
> Best regards,
>
> Authors

---

### Note · Authors · 2026-01-01

I have read and agree with the venue's withdrawal policy on behalf of myself and my co-authors.